# Heterogeneity of multiple sclerosis lesions in fast diffusional kurtosis imaging

Christian Thaler[1]*, Anna A. Kyselyova[1], Tobias D. Faizy[1], Marie T. Nawka[1], Sune Jespersen[2], Brian Hansen[2], Jan-Patrick Stellmann[3,4,5,6], Christoph Heesen[3,4], Klarissa H. Stürner[3,4,7], Maria Stark[8], Jens Fiehler[1], Maxim Bester[1ᵒ], Susanne Gellißen[1ᵒ]

**1** Department of Diagnostic and Interventional Neuroradiology, University Medical Center Hamburg-Eppendorf, Hamburg, Germany, **2** Department of Clinical Medicine - Center of Functionally Integrative Neuroscience, Aarhus University, Aarhus, Denmark, **3** Department of Neurology, University Medical Center Hamburg-Eppendorf, Hamburg, Germany, **4** Institute for Neuroimmunology and Clinical MS Research, University Medical Center Hamburg-Eppendorf, Hamburg, Germany, **5** APHM, Hospital de la Timone, CEMEREM, Marseille, France, **6** Aix Marseille University, CNRS, CRMBM, UMR 7339, Marseille, France, **7** Department of Neurology, University Hospital Schleswig-Holstein, Kiel, Germany, **8** Institute of Medical Biometry and Epidemiology, University Medical Center Hamburg-Eppendorf, Hamburg, Germany

ᵒ These authors contributed equally to this work.
* c.thaler@uke.de

**Data Availability Statement:** All relevant data are within the paper and its Supporting information file.

**Funding:** The author(s) received no specific funding for this work.

## Abstract

### Background

Mean kurtosis (MK), one of the parameters derived from diffusion kurtosis imaging (DKI), has shown increased sensitivity to tissue microstructure damage in several neurological disorders.

### Methods

Thirty-seven patients with relapsing-remitting MS and eleven healthy controls (HC) received brain imaging on a 3T MR scanner, including a fast DKI sequence. MK and mean diffusivity (MD) were measured in the white matter of HC, normal-appearing white matter (NAWM) of MS patients, contrast-enhancing lesions (CE-L), FLAIR lesions (FLAIR-L) and black holes (BH).

### Results

Overall 1529 lesions were analyzed, including 30 CE-L, 832 FLAIR-L and 667 BH. Highest MK values were obtained in the white matter of HC (0.814 ± 0.129), followed by NAWM (0.724 ± 0.137), CE-L (0.619 ± 0.096), FLAIR-L (0.565 ± 0.123) and BH (0.549 ± 0.12). Lowest MD values were obtained in the white matter of HC (0.747 ± 0.068 $10^{-3}$mm²/sec), followed by NAWM (0.808 ± 0.163 $10^{-3}$mm²/sec), CE-L (0.853 ± 0.211 $10^{-3}$mm²/sec), BH (0.957 ± 0.304 $10^{-3}$mm²/sec) and FLAIR-L (0.976 ± 0.35 $10^{-3}$mm²/sec). While MK differed significantly between CE-L and non-enhancing lesions, MD did not.

### Conclusion

MK adds predictive value to differentiate between MS lesions and might provide further information about diffuse white matter injury and lesion microstructure.

**Competing interests:** The authors have declared that no competing interests exist.

## Introduction

Multiple sclerosis (MS) is a chronic inflammatory disease of the central nervous system, presenting a broad spectrum of histopathological processes, such as inflammation, demyelination, axonal loss and the formation of astrocytic scars [1–3]. To investigate and evaluate this pathological heterogeneity, MRI has become a well-established tool in research and clinical routine. While conventional MRI techniques are mostly limited to quantify lesion count, lesion volume and the detection of contrast-enhancing lesions (CE-L), quantitative MRI techniques allow the detection of diffuse white and gray matter injury [4–6]. Furthermore, advanced techniques such as diffusion tensor imaging (DTI) have shown promising results to identify blood-brain-barrier breakdown in lesions without the application of gadolinium [7, 8]. Conventional DTI approximates diffusion displacement with a Gaussian distribution, which is accurate for long diffusion times and small diffusion weighting (see [9] for details), or when diffusion occurs in a free and unrestricted environment. In brain tissue however, diffusion of water molecules is restricted by numerous components, such as complex underlying cellular components and structures, which are not adequately characterized by DTI. To obtain better sensitivity to such microstructure, DKI has been introduced as a complementary technique that incorporates non-Gaussian diffusion effects [10, 11]. Mean kurtosis (MK), one of the parameters derived from DKI, has shown increased sensitivity to tissue microstructure in several neurological disorders such as stroke, head trauma, glioma and neurodegenerative diseases [12–15]. A decreased MK has been associated with lower axonal and myelin density, giving further insights into MS lesions' histopathology [16, 17].

Just recently, this research group demonstrated the feasibility to assess MS lesion quantitatively using quantitative imaging techniques, such as myelin water imaging (MWI) [18]. Hereby, we were able to differentiate visually assigned MS lesions and addtionally, evaluate lesional damage. The aim of this current study is to apply DKI in a cohort of MS patients and evaluate lesional values derived from this technique. We hypothesize that the derived parameters will help to differentiate MS lesion types according to tissue damage and furthermore, might even give additional information about blood-brain-barrier breakdown, i.e. be able to identify contrast enhancing lesions.

## Methods

### Study cohort

Thirty-seven patients with relapsing-remitting MS and eleven age-matched healthy controls were enrolled in this prospective monocentric study from February 2014 to November 2014. Inclusion criteria were as follows: age 18–70 years, diagnosis of relapsing-remitting MS according to the 2010 revised McDonald criteria [19] and absence of neurologic conditions other than MS. Patients with primary progressive MS or patients with contraindications to contrast agent were excluded. Also, all patients were treated dimethyl fumarate prior to this study. Inclusion criteria of the aged matched-healthy controls was the absence of any thrombotic, vascular, or neurological disease. An overview over the study population is given in Table 1. The study was approved by the local Ethics Committee Hamburg (Ethik-Kommission der Ärztekammer Hamburg) following the guidelines of the Declaration of Helsinki and all subjects provided written informed consent.

### MRI data acquisition

All MR scans were performed on a 3 Tesla MR scanner (Magnetom Skyra, Siemens Healthcare GmbH, Erlangen, Germany). The MR protocol included a sagittal 3-dimensional FLAIR

**Table 1. Subject population.**

|  | MS patients | Healthy controls |
| --- | --- | --- |
| Median age (in years) | 36.2 (range: 18.6–64.9) | 34.8 (range: 23.5–59.1) |
| Sex | 24 female: 13 male | 9 female: 2 male |
| Disease course | 37 RRMS | n.a. |
| Median disease duration (in years) | 4 (range: 4–22 years) | n.a. |
| Median EDSS | 2 (range: 0–5.5) | n.a. |

EDSS = Expanded Disability Status Scale, RRMS = relapsing-remitting Multiple Sclerosis.

(echo-time (TE) = 390 ms, repetition time (TR) = 4700 ms, inversion time (TI) = 1800 ms, 192 slices, field of view (FOV) = 256 mm, voxel size = 1.0 x 1.0 x 1.0 mm) and a T1-weighted magnetization-prepared rapid gradient echo (MPRAGE) sequence before and after Gadolinium injection (TE = 2.43 ms, TR = 1900 ms, TI = 900 ms, 192 slices, FOV = 256 mm, voxel size = 1.0 x 1.0 x 1.0 mm, flip-angle = 9˚). For the DKI data acquisition, a fast kurtosis protocol, introduced by Hansen et al. [20, 21], was applied before Gadolinium administration (TE = 100 ms, TR = 4300 ms, 60 slices, FOV = 196 mm, acquisitions = 13 (one along each of the x-, y- and z-directions) with $b$ = 1000 s/mm$^2$, 9 different directions with $b$ = 2500 s/mm$^2$, and 1 acquisition with $b$ = 0 s/mm$^2$).

## DKI processing

Prior to processing data was inspected visually to ensure that data was free of artifacts (see also discussion for more on data quality assurance). Following this, data was preprocessed to correct for eddy current distortions and bulk motion using the "eddy" function from the FMRIB Software Library (approx.. 30 min) (FSL, http://fsl.fmrib.ox.ac.uk/fsl/fslwiki/) [22].

Further processing of DKI data was performed using an in-house developed software (https://github.com/sunenj/Fast-diffusion-kurtosis-imaging-DKI) running in Matlab 7 (Mathworks, Sherborn, MA, USA), which has been validated and reported before [23]. Postprocessing steps included denoising, Rice-floor adjustment and correction for Gibbs ringing effects (approx.. 1 minute) [24–26]. Subsequently, parameter maps for DKI-derived indices of MK and mean diffusivity (MD) were obtained for each patient. Overall processing time was 3–5 seconds per patient. The fast DKI protocol provides a MK variant sometimes referred to as the tensor-derived mean kurtosis or mean (of the) kurtosis tensor (MKT). This metric has been shown to be very similar to the conventional MK but has the advantage that it can be measured and calculated very rapidly [27]. In short, the diffusion tensor trace is invariant and can be assessed from three orthogonal directions (each yielding a diffusivity Dxx, Dyy, and Dzz). MD is then calculated as MD = (Dxx+Dyy+Dzz)/3. In the 1-3-9 scheme employed here, the MD estimated this way takes into account kurtosis by also using the high b-value data for the x, y, and z directions in the MD estimate. Details are given in [20] where it was shown that this estimation scheme is very accurate when compared to DKI derived MD estimates based on much larger data sets. An example of the acquired images and corresponding MK and MD parameter maps is displayed in Fig 1.

## Image analysis

In MS patients, we defined four different areas of white matter damage, which included: normal-appearing white matter (NAWM), contrast-enhancing lesions (CE-L), T2/FLAIR-hyperintense lesions (FLAIR-L) and black holes (BH).

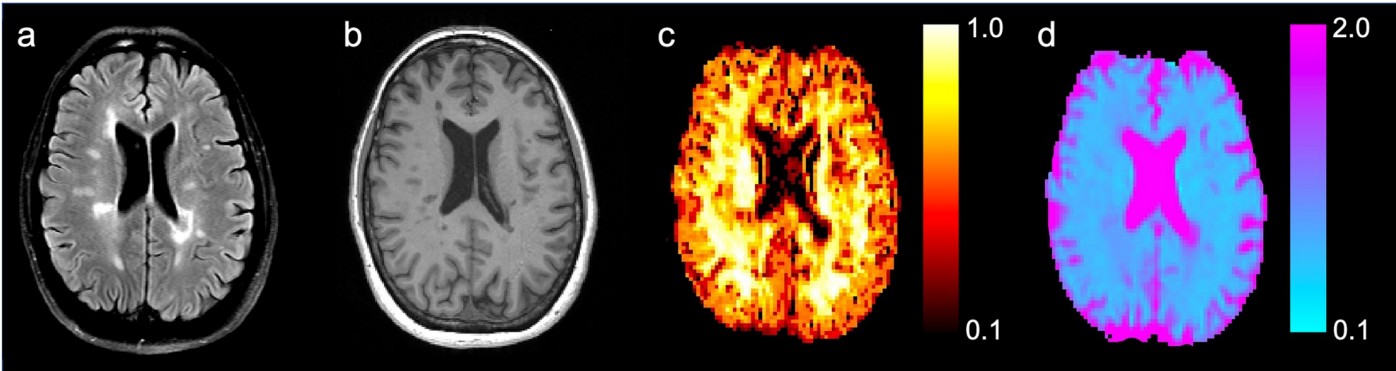

**Fig 1. Example of an MR scan of an MS patient.** Hyperintense FLAIR lesions displayed on FLAIR images (a) and unenhanced T1w images (b) with the corresponding colour-coded mean kurtosis (MK) (c) and mean diffusivity (MD) (in$10^{-3}$mm$^2$/sec) (d) parameter maps.

For NAWM, we outlined six regions of interest (ROI) in the occipital, frontal and parietal white matter on FLAIR images. ROIs were placed manually by two neuroradiologists (C.T. and T.F.) both with 5 years of experience in MS imaging and in consideration of the LST segmentations to avoid partial volume errors from MS lesions.

CE-L were manually outlined by the same two neuroradiologists by consensus on T1w-MPRAGE images after contrast using the segmentation software Analyze 11.0 (AnalyzeDirect, Inc. KS, USA). In case a lesion showed a ringlike enhancement pattern, only parts of the lesion that were within the ring enhancement were segmented. In case a lesion showed a nodular enhancement pattern, only the nodular enhancing parts were segmented. The resulting CE-L masks of both raters were binarized and patient-wise multiplied to obtain a consensus mask.

BH were defined as non-enhancing lesions that appear hypointense on T1w-images with signal intensity below cortex and are concordant with hyperintense lesions on a T2w-image. BH were marked applying the same algorithm as described for CE lesion outlining.

FLAIR-L were semi-automatically segmented on FLAIR images using an open source lesion segmentation software (LST: Lesion Segmentation Tool) [28]. All FLAIR-L were controlled for accuracy and lesion outlining was manually corrected if necessary. Since CE-L and BH are also hyperintense in T2w images, they were subtracted from the FLAIR-L mask, leaving only those regions visible on FLAIR/T2w images that did not correspond to a CE-L or BH. To minimize partial volume effects, lesions smaller than 10 voxels on FLAIR images were excluded from final data analysis.

Additionally, six ROIs were placed in the occipital, frontal and parietal white matter of the healthy controls (HC), using the same workflow as described above for NAWM. An example of lesion outlining and ROI placement is given in Fig 2.

## Image registration

FLAIR images were registered to T1w-MPRAGE images using FMRIB's Linear Image Registration Tool (approx.. 2 minutes) (FLIRT; FSL) [29]. Subsequently, the FLAIR-L masks and NAWM ROIs, both on FLAIR images, were transformed into the T1w-MPRAGE space using the resulting transformation matrices. To co-register DKI with the T1w-MPRAGE images, the b = 0 images were registered to the T1w space using FSL's epi_reg (approx.. 60 minutes). The obtained transformation matrices were inverted and subsequently applied to the lesion masks. All transformed lesions masks, now in DKI space, were visually confirmed for quality control.

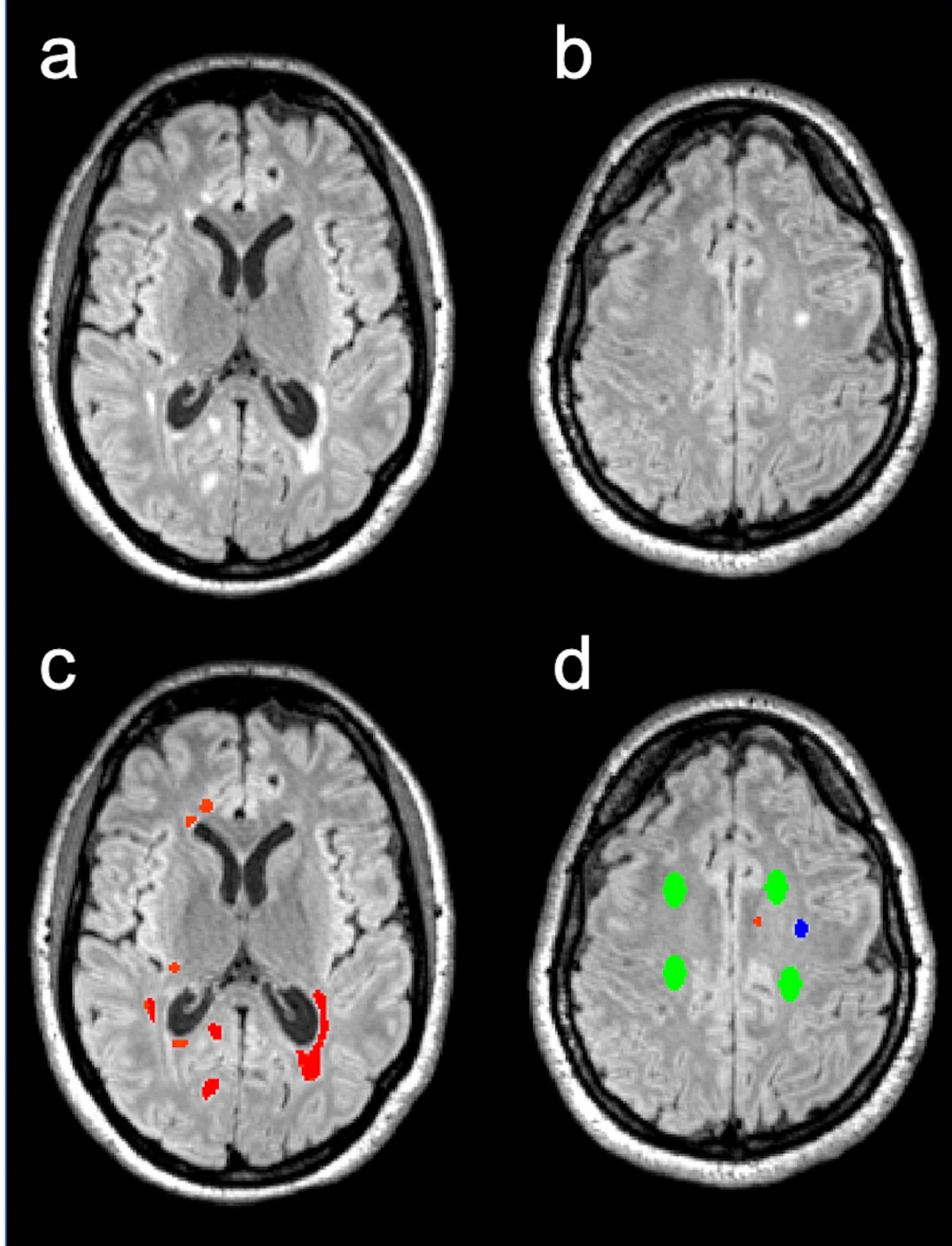

**Fig 2. Example of lesion outlining and ROI placement.** FLAIR images of an MS patient before lesion outlining (a, b) and after lesion outlining (c, d). FLAIR lesion = red, black holes = blue, NAWM = green.

Finally, the binarized lesion masks were multiplied with the corresponding MD and MK maps to obtain mean MD and MK values for each lesion and ROI.

## Statistical analysis

Statistical analysis was performed using SPSS IBM SPSS 21.0. (IBM Corp., Armonk, NY, USA). To improve comparability between the measures and adjust figure scales, MD values are displayed multiplied with the factor 100. To compare means of MK and MD between the different lesion categories we used the Kruskal–Wallis one-way analysis of variance. Subsequently, Dunn's pairwise test was carried out for group analysis. Additionally, p-values were adjusted using the Bonferroni error correction and two-sided p-values of less than 0.05 were considered as statistically significant. Additionally, we repeated the analysis but combined the results from FLAIR-L and BH into one group, i.e. non-enhancing lesions (NE-L).

To avoid potential bias caused by the uneven distribution of MS lesions in our study cohort, a multinomial logistic regression model was applied to calculate the odds ratios to observe a certain lesion type in dependence of the values of MD and MK. Only the lesion categories NAWM, CEL, BH and FLAIR-L were used. FLAIR-L were determined as the reference category. The lesion categories represented the dependent variable, while MK and MD were the predictors. The influence of MD and MK was adjusted for the age, the disease duration, EDSS and the lesion size by adding them as independent variables into the model. All independent variables were considered as person-specific characteristics in the model whereby the dependence of several measures resulting from one patient was respected. 95%-confidence intervals and p-values for the odds ratios are reported.

Also, Pearson's correlations coefficients between MK and MD for each lesion group were calculated. Again, a two-sided p-value of less than 0.05 was considered as statistically significant.

## Results

Overall 1529 lesions were detected and analyzed, including 30 CE-L, 832 FLAIR-L and 667 BH. CE-L were detected in 11 patients. Mean lesion count per patient was 41.3 lesions, ranging from 2 to 118 lesions. Inter-rater reliability for the outlining of BH was good with a Dice coefficient of 0.8.

## MK analysis

MK values for each lesion type and ROI are displayed in Fig 3. In summary, highest MK values were obtained in the white matter of HC (0.814 ± 0.129), followed by NAWM in MS patient (0.724 ± 0.137), CE-L (0.619 ± 0.096), FLAIR-L (0.565 ± 0.123) and BH (0.549 ± 0.12). Kruskal–Wallis one-way analysis of variance was significant for MK ($p<0.001$). Using Dunn's pairwise test, we found significant differences for MK between most lesion categories, which is displayed in Table 2. No significant differences were found between WM of HC and NAWM (p = 0.052), CE-L and FLAIR-L (p = 0.133) and FLAIR-L and BH (p = 0.363). Additionally, we repeated the Kruskal–Wallis one-way analysis of variance and Dunn's pairwise test, but replaced FLAIR-L and BH with NE-L. By doing so, we detected a significant difference between CE-L (0.619 ± 0.096) and NE-L (0.558 ± 0.122) (p = 0.035).

## MD analysis

MD values for each lesion type and ROI are displayed in Fig 4. Lowest MD values were obtained in the white matter of HC (0.747 ± 0.068 $10^{-3} mm^2/sec$), followed by NAWM in MS patient (0.808 ± 0.163 $10^{-3} mm^2/sec$), CE-L (0.853 ± 0.211 $10^{-3} mm^2/sec$), BH (0.957 ± 0.304

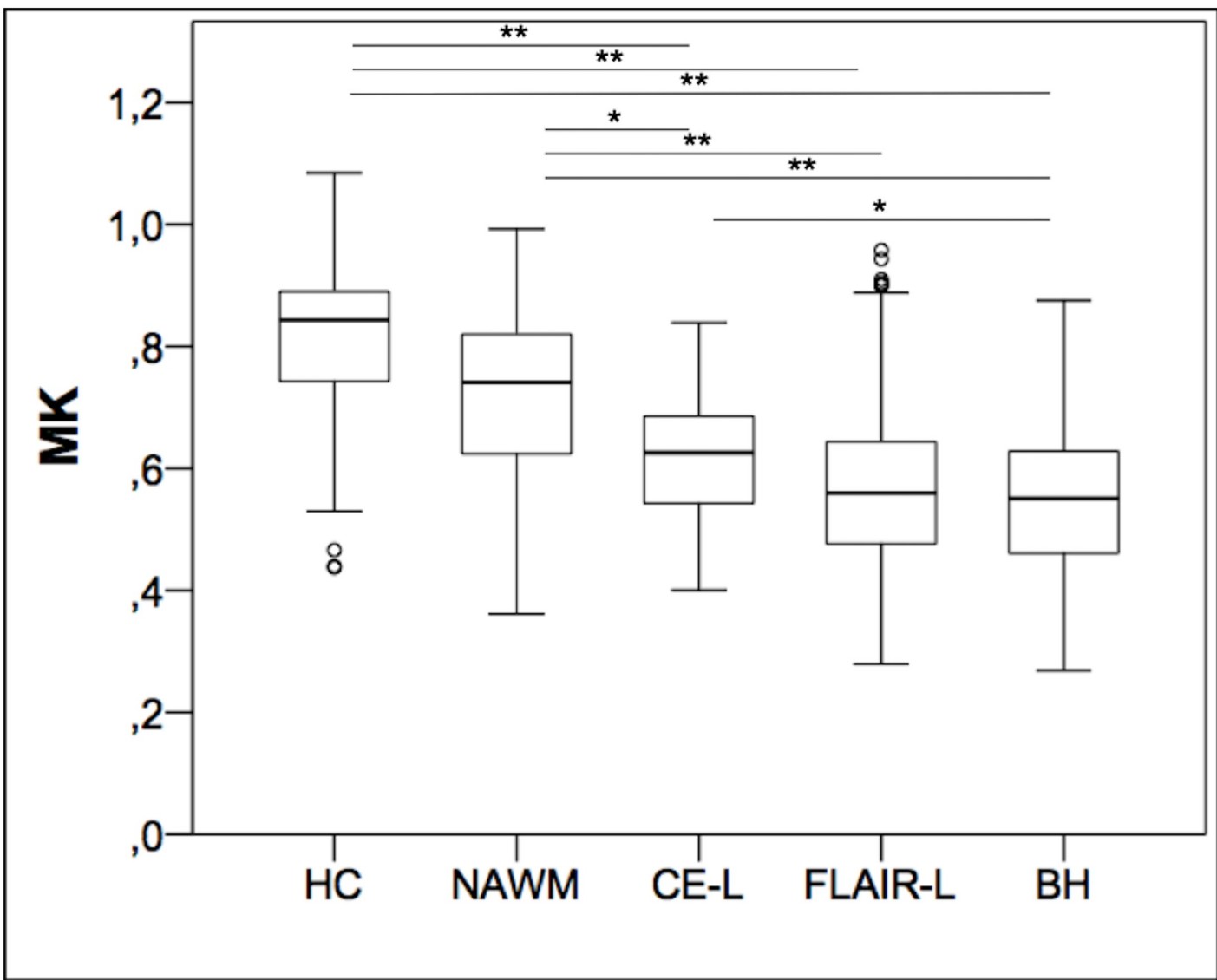

**Fig 3. Distribution of mean kurtosis (MK) within the different lesion categories.** Boxplots of MK in the white matter of healthy controls (HC), normal appearing white matter (NAWM) of MS patients, contrast-enhancing lesions (CE-L), FLAIR lesions (FLAIR-L) and black holes (BH). * = p<0.05, ** = p<0.001.

$10^{-3}$mm$^2$/sec) and FLAIR-L (0.976 ± 0.35 $10^{-3}$mm$^2$/sec). Again, Kruskal–Wallis one-way analysis of variance was significant for MD (p<0.001). However, using Dunn's pairwise test, no significant differences were found between WM of HC and NAWM (p = 0.1), NAWM and CE-L (p = 1.00), CE-L and FLAIR-L (p = 0.24), CE-L and BH (p = 0.16) and FLAIR-L and BH (p = 1.00). Even after combining FLAIR-L and BH to NE-L and repeating Kruskal–Wallis one-way analysis of variance and Dunn's pairwise test, we could not find a significant difference between CE-L (0.853 ± 0.211 $10^{-3}$mm$^2$/sec) and NE-L (0.968 ± 0.33 $10^{-3}$mm$^2$/sec) (p = 0.11).

## Multinomial logistic regression

The odds ratios give the odds to develop the certain lesion type compared to the reference level (FLAIR-L) if MK or MD increases by one unit, respectively. For MK, we found significant

**Table 2. Differences between MS lesion categories regarding MK and MD.**

| | MK | | MD | |
|---|---|---|---|---|
| | p-value | Adjusted p-value | p-value | Adjusted p-value |
| HC—NAWM | 0.005 | 0.052 | 0.01 | 0.1 |
| HC—CE-L | <0.001 | **<0.001** | 0.002 | **0.015** |
| HC—FLAIR-L | <0.001 | **<0.001** | <0.001 | **<0.001** |
| HC—BH | <0.001 | **<0.001** | <0.001 | **<0.001** |
| NAWM—CE-L | 0.004 | **0.039** | 0.111 | 1.0 |
| NAWM—FLAIR-L | <0.001 | **<0.001** | <0.001 | **<0.001** |
| NAWM—BH | <0.001 | **<0.001** | <0.001 | **<0.001** |
| CE-L—FLAIR-L | 0.013 | 0.133 | 0.024 | 0.241 |
| CE-L—BH | 0.002 | **0.023** | 0.016 | 0.156 |
| FLAIR-L—BH | 0.036 | 0.363 | 0.539 | 1.0 |

Kruskal–Wallis one-way analysis of variance and Dunn's pairwise test was carried out to test for differences between the groups. Additionally, p-values were adjusted using the Bonferroni error correction. HC = healthy cohorts, NAWM = normal-appearing white matter, CE-L = contrast-enhancing lesion, FLAIR-L = FLAIR lesion, BH = black hole, MK = mean kurtosis, MD = mean diffusivity.

odds ratios for every lesion type. For MD, we could only obtain a significant odds ratio for NAWM. All odds ratios, confidence intervals and p-values are displayed in Tables 3 and 4.

## Correlation between MK and MD

We only obtained moderate correlations between MK and MD within any ROI or lesion category. Strongest correlations were found in NAWM in MS patients with a Pearson's correlation coefficient of -0.47 ($p<0.001$), followed by CE-L (r = -0.42; p = 0.02), white matter of HC (r = -0.35; p = 0.002), FLAIR-L (r = -0.275; $p<0.001$) and BH (r = -0.203; $p<0.001$). We calculated an overall correlation coefficient of -0.306 ($p<0.001$).

## Discussion

There is a growing number of studies using quantitative MRI, such as DTI, magnetization transfer imaging (MTI) or quantitative susceptibility mapping (QSM), to differentiate between lesion stages in MS patients. In a meta-analysis by Gupta et al. [7], only FA and QSM have shown reasonable accuracy measures to differentiate between enhancing and non-enhancing lesions in MS patients. Especially QSM seems to be a sensitive imaging marker to predict lesion enhancement [30–32]. However, studies focusing on potential and promising imaging markers in MS are still rare, implicating a need for additional quantitative measures to evaluate white matter and lesional damage. In this study, we applied a recently introduced fast DKI sequence to characterize different visually assigned MS lesion types. Though DKI has been applied to detect diffuse gray and white matter injuries in MS patients, this is the first study to characterize DKI properties in different MS lesion types [33–38]. In our cohort, MK, one of the parameters arrived from the fast DKI sequence, differed significantly between most lesion categories, suggesting a good discriminative value for classifying white matter damage in MS patients. These findings are supported by an additional multinomial logistic regression model, that compared the different lesion categories in MS patients and included patient specific characteristics such as the patients' age, disease duration, EDSS and lesion size as independent variables. Most importantly, MK values within CE-L were significantly higher than MK values within NE-L (CE-L = 0.619 vs. NE-L = 0.558; p = 0.035). However, we did not find significant

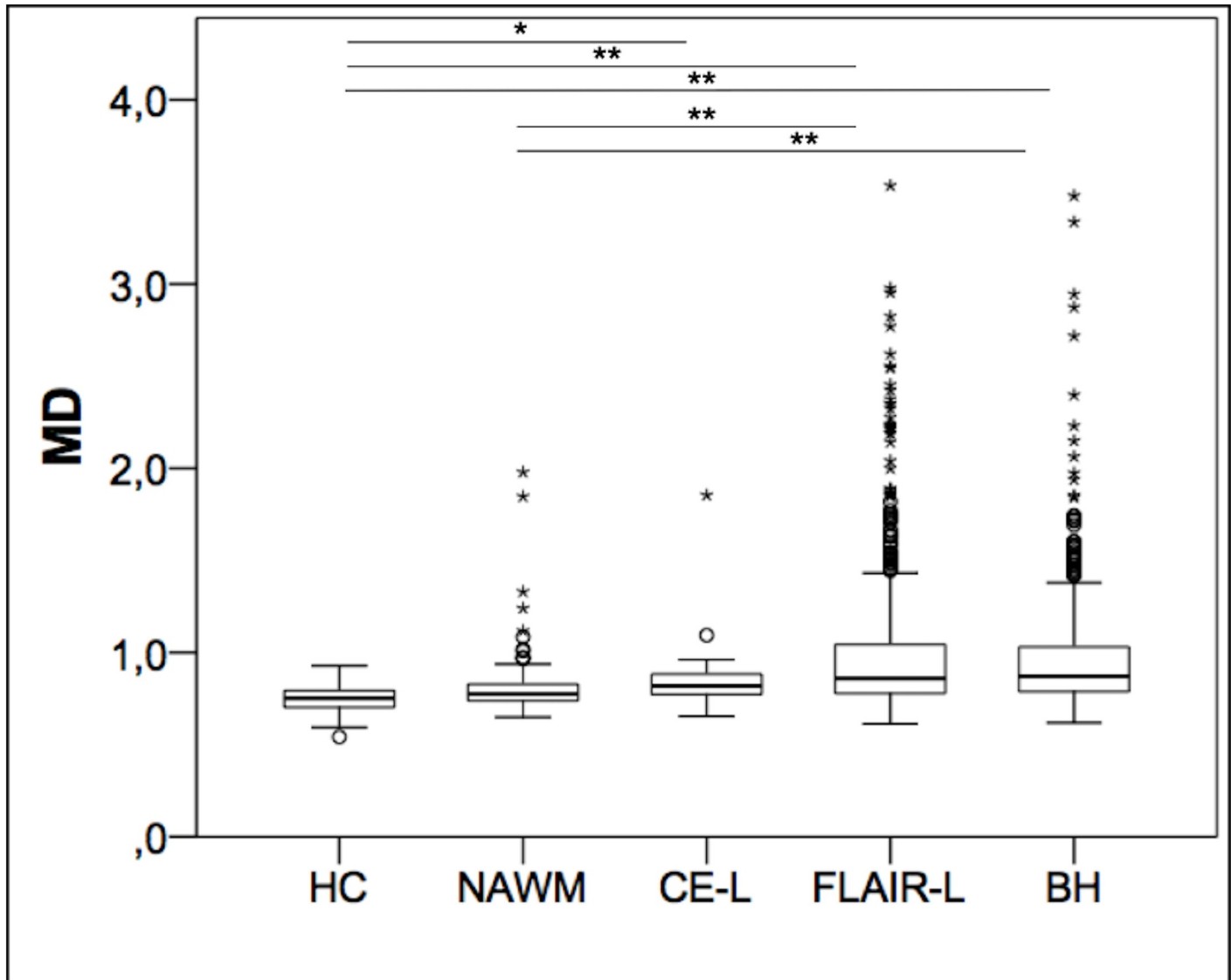

**Fig 4. Distribution of mean diffusivity (MD) within the different lesion categories.** Boxplots of MD ($10^{-3}$mm$^2$/sec) in the white matter of healthy controls (HC), normal appearing white matter (NAWM) of MS patients, contrast-enhancing lesions (CE-L), FLAIR lesions (FLAIR-L) and black holes (BH). * = p<0.05, ** = p<0.001.

**Table 3. Odds ratios MK.**

| MK | Odds Ratio | Lower bound 95% CI | Upper bound 95% CI | p |
|---|---|---|---|---|
| NAWM | 100.17*$10^3$ | 10.68*$10^3$ | 800.15*$10^3$ | **<0.001** |
| CE-L | 5.03 | 1.82 | 1.52*$10^3$ | **<0.05** |
| BH | 0.191 | 0.067 | 0.54 | **<0.01** |

Odds Ratios with the lower and upper bound of the 95%-confidence intervals and the p-values for mean kurtosis (MK) in normal-appearing white matter (NAWM), contrast enhancing lesions (CE-L) and black holes (BH) compared to FLAIR lesions (FLAIR-L) as reference category. Multinomial regression also included age, disease duration, EDSS and lesion size.

**Table 4. Odds ratios MD.**

| MD | Odds Ratio | Lower bound 95% CI | Upper bound 95% CI | p |
|------|------------|--------------------|--------------------|--------|
| NAWM | 0.014 | 0.003 | 0.059 | <**0.001** |
| CE-L | 0.227 | 0.033 | 1.559 | 0.13 |
| BH | 1.013 | 0.759 | 1.353 | 0.93 |

Odds Ratios with the lower and upper bound of the 95%-confidence intervals and the p-values for mean diffusivity (MD) in normal-appearing white matter (NAWM), contrast enhancing lesions (CE-L) and black holes (BH) compared to FLAIR lesions (FLAIR-L) as reference category. Multinomial regression also included age, disease duration, EDSS and lesion size.

differences for MD between NAWM and CE-L or between CE-L and NE-L, which is in line with a recent meta-analysis by Gupta et al [7]. While CE-L are associated with a breakdown of the blood-brain-barrier, indicating active inflammation with immigration of inflammatory cells and beginning demyelination, FLAIR-L and BH display usually lesions in an advanced or chronic state and are proposed as a more specific marker for axonal loss and parenchymal destruction [1, 2, 39]. MK is associated with tissue complexity and heterogeneity, depending on cell membrane density, orientation and organization of fibers [40, 41]. Furthermore, it has been reported that a decreased MK is associated with diffuse white matter damage and axonal degeneration in MS patients [36]. Therefore, MK seems to be potentially a more sensitive marker than DKI-derived index of MD for these tissue heterogeneities and white matter damage, that are found in different stages of MS lesion formation. Most importantly, MK could be used as additional imaging marker to evaluate lesion acuity without the use of gadolinium.

We could not find significant differences between MK within the white matter of HC and NAWM of MS patients, though a trend towards decreased MK within the NAWM of MS patients was observed (p = 0.052). Looking at recent studies, results are varying. Li et al. and Qian et al. [33, 36] found significant differences between MK within the white matter of HC and NAWM of MS patients, but Bester et al. [38] did not. A mean disease duration of 6.0 years in our study cohort is in range with these former studies (Qian et al.: 7.9 years; Li et al.: 4.5 years; Bester et al.: 3.5 years) and seems unlikely to be the cause of the diverging results. Still, potential differences in our study cohort compared to the study cohorts from the studies mentioned above, such as the treatment with dimethyl fumarate or other uncharacterized parameters, might explain our results. Further, it has to be stated that MK is an empirical diffusion measure and offers low microstructural and pathological specificity to evaluate tissue damage in NAWM [11]. Also, while we used a manually drawn ROI based approach to select regions of NAWM, the studies mentioned before used tract based spatial statistics or used ROI based analysis including the whole white matter, which might account for the conflicting results.

We used a novel and fast DKI sequence introduced by Hansen et al., which has been applied to detect white matter and tissue alterations in different neurological conditions [15, 41–45]. While scan durations for the DKI sequences used in similar studies were between 8 and 19 minutes (depending on the b-values), our DKI scan time was only 2 minutes 46 seconds [33, 34, 36, 38]. Postprocessing of the DKI data was performed with an in-house algorithm implemented in Matlab with a processing time of 3–5 seconds. With its very fast acquisition time and post-processing steps, it can easily be implemented in clinical routine or MS study protocols. The fast DKI method provides an MK metric that differs slightly from conventional MK in its definition [20, 21, 27]. Briefly, the fast kurtosis schemes estimate a mean kurtosis variant calculated as one fifth of the kurtosis tensor trace, often referred to as the MKT (MK in this study). This metric can be rapidly and robustly estimated from few, distinct encoding

directions as done here. From a theoretical perspective the MKT is identical to the conventional MK only in isotropic media [10]. In extensive comparisons between the MKT and the conventional MK are made and it was found that MK and MKT are very similar in all but the most anisotropic tissues where differences of the order of approximately 5% are seen in WM with MK>1 [20]. This was confirmed in both in ex vivo examinations and in vivo human brain. We therefore believe that MKT is a very good surrogate marker for MK even in highly anisotropic WM and that the results from the present study can be said to be generally indicative of the value of DKI in MS. Viewed in isolation, the results of the present study are very promising for the use of fast DKI in MS diagnostics.

As the use of diffusion MRI (dMRI) for diagnostics increases, much attention has been given to data dMRI data quality assurance e.g. Malyarenko et al. [46], Fedeli et al. [47]. This is also very relevant to the fast DKI methods which employ distinct diffusion encoding directions input to the scanner in a custom gradient table. While our pre-processing takes into account image distortions due to e.g. eddy currents (see methods) more subtle effects may influence dMRI encoding where effective encoding directions may deviate from the prescribed ones due gradient non-linearities and eddy currents potentially affection both encoding directions and strength (b-value) with the consequence that effective encoding shells become uneven (rugged). The effect of this on the fast DKI estimates was evaluated extensively by Hansen et al. [21] and the schemes were found to be very robust to all but the most extreme encoding deviations. Nevertheless, when including data from multiple scan system in a study a dMRI encoding details may vary greatly depending on vendor and sequence is needed [47]. In such cases a stricter scheme for verifying data quality and encoding fidelity is needed as proposed in [48].

There are several limitations to this study. The main limitation of our study is the uneven distribution of lesions in the different groups. While we were able to include 832 FLAIR-L and 667 BH, only 30 CE-L were detected in our cohort. However, this is a common distribution in MS patients, since contrast-enhancement is a sporadic phenomenon in MS and can even be completely absent on MRI. This was taken into account by our statistical analysis, and still significant differences were obtained for MK. We were able to include only eleven healthy controls who were age-matched, but not matched for sex. It was of high importance to select an aged-matched control group since age can significantly influence MK and MD values within the white matter, while the impact of sex on DTI parameters in the white matter, especially in younger populations, is still controversial [49–51]. So far, studies focusing on sex-specific differences of DKI-derived indices in the white matter are missing, which has to be taken into consideration when interpreting our results. Further, we like to emphasize that this study was not a case-control-study to investigate differences between MS patients and healthy controls. Healthy controls were merely included to obtain a reference values for MK and MD. We are also aware that our classification does not reflect the current histological definition of MS lesion types as proposed by Kuhlmann et al. [52]. However, since histological correlation was not available for this study group, we were dependent on a visual grading to classify MS lesions which might not take the broad variability of histopathological processes within a lesion into account. Further, our DK images offered lower resolution (voxel size = 2 x 2 x 2 mm$^3$) compared to the T2w-FLAIR and MPRAGE images (voxel size = 1 x 1 x 1 mm$^3$), in which lesion outlining was performed. Also, we did not obtain a reversed phase-encoding direction b = 0 image and could not correct for susceptibility induced distortions. We used FSL's epi_reg to register the b0 image to the T1w space and image co-registrations were checked carefully to exclude inaccurate registrations. Still, minor co-registration errors were possible. Image artifacts on the border of the ventricles may influence values of lesions within this location which might explain increased MD values in some MS lesions. To prevent major partial volume effects and negative or aberrant values within certain areas, image co-registrations and lesion

ROIs in the DKI space were also checked carefully and excluded or manually corrected if necessary. Finally, we did not compare MK and MD acquired by the fast DKI protocol with conventional Gaussian DTI-derived measures. The reason for this is that in our study, the focus is on the use of fast DKI in MS, and therefore the MD we report is the DKI-derived MD. It is worth noting that the estimates of MD derived from DKI can differ from MD estimates from DTI [20, 53, 54]. This is expected, since the diffusion signal at b ~ 1000s/mm$^2$ can still be affected by nongaussian diffusion, which in DTI must be captured by the only adjustable parameter, i.e. the diffusion tensor. In DKI, these effects are appropriately absorbed by the kurtosis tensor, which is why the MD estimate from DKI is a better estimate of the true MD [55]. For this reason, the DKI framework is recommended over DTI when scan times allow. With the fast DKI protocols this is almost always feasible. On the other hand, the fast DKI protocol offers less robust estimates of MK compared to a full DKI protocol. Nevertheless, the estimation scheme used in this study has been shown to be very accurate when compared to estimates based on much larger data sets [20].

Finally, we note that the diffusion gradients were not calibrated prior to data acquisition. Thus, the actual gradient valuables may differ from the nominal b-values, leading to a systematic bias in our results [47, 48]. However, while it may affect absolute parameter values, we do not expect it to affect the findings of group differences.

## Conclusion

In conclusion, MK offered good discriminative value between different MS lesion types. Most importantly, MK differed significantly between enhancing and non-enhancing MS lesions, supporting its usefulness as a potentially image marker to assess lesion acuity without the use of gadolinium, which is in general a point of critical debate [56, 57]. With the fast acquired and easy to post-process DKI sequence used in this study, it can be easily implemented into research protocols and most importantly, clinical routine. Still, further research is needed to replicate and support our findings. It would be highly interesting to apply DKI in combination with other advanced quantitative MR sequences, such as MWI, QSM or alternative diffusion imaging schemes, such as neurite orientation dispersion and density imaging (NODDI) and spherical mean technique (SMT), which have shown very promising results to evaluate white lesional damage in MS patients [18, 32, 58]. Of course, histopathological correlations would provide the best evidence to verify our findings but are naturally limited.

## Supporting information

**S1 Data.**
(XLS)

## Author Contributions

**Conceptualization:** Christian Thaler, Brian Hansen, Jens Fiehler, Maxim Bester, Susanne Gellißen.

**Data curation:** Christian Thaler, Anna A. Kyselyova, Tobias D. Faizy, Jan-Patrick Stellmann, Christoph Heesen, Klarissa H. Stürner, Susanne Gellißen.

**Formal analysis:** Christian Thaler, Tobias D. Faizy, Sune Jespersen.

**Funding acquisition:** Jan-Patrick Stellmann, Christoph Heesen, Jens Fiehler.

**Investigation:** Christian Thaler, Anna A. Kyselyova, Tobias D. Faizy, Marie T. Nawka, Sune Jespersen, Maxim Bester, Susanne Gellißen.

**Methodology:** Christian Thaler, Anna A. Kyselyova, Sune Jespersen, Brian Hansen, Klarissa H. Stürner, Maria Stark, Susanne Gellißen.

**Project administration:** Jens Fiehler, Susanne Gellißen.

**Resources:** Sune Jespersen, Brian Hansen, Jan-Patrick Stellmann, Jens Fiehler.

**Software:** Sune Jespersen, Brian Hansen.

**Supervision:** Christian Thaler, Sune Jespersen, Maria Stark, Jens Fiehler, Maxim Bester, Susanne Gellißen.

**Validation:** Christian Thaler, Tobias D. Faizy, Marie T. Nawka, Sune Jespersen, Brian Hansen, Maria Stark, Maxim Bester.

**Visualization:** Christian Thaler.

**Writing – original draft:** Christian Thaler.

**Writing – review & editing:** Christian Thaler, Anna A. Kyselyova, Tobias D. Faizy, Marie T. Nawka, Sune Jespersen, Brian Hansen, Jan-Patrick Stellmann, Christoph Heesen, Klarissa H. Stürner, Maria Stark, Jens Fiehler, Maxim Bester, Susanne Gellißen.

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
