## [Decision Letter · Decision Letter 0]

24 Nov 2020

PONE-D-20-30787

Heterogeneity of Multiple Sclerosis Lesions in Fast Diffusional Kurtosis Imaging

PLOS ONE

Dear Dr. Thaler,

Thank you for submitting your manuscript to PLOS ONE. After careful consideration, we feel that it has merit but does not fully meet PLOS ONE’s publication criteria as it currently stands. Therefore, we invite you to submit a revised version of the manuscript that addresses the points raised during the review process.

In addition to the comments raised by Reviewer 2, please also address the following:

1. The word “gender” is used in the manuscript, although “sex” is also used. Usually, “sex” (the biological designation) is meant. “Gender” is the social construct and is rarely relevant in neurologic disease. Please revise the text to use “sex” rather than “gender” throughout.

2. It is stated that the DKI acquisition is very quick and that the post-processing requires only a few seconds. However, this does not account for other aspects of the pre-processing (e.g. use of eddy, co-registration, etc.) As you have included exact timings for the acquisition and Matlab processing, please include timings for these other vital steps as well.

3. Is that Matlab code for the DKI processing being made publicly available as well?

We look forward to receiving your revised manuscript.

Kind regards,

Niels Bergsland

Academic Editor

PLOS ONE

Journal Requirements:

2. We note you have included a table to which you do not refer in the text of your manuscript. Please ensure that you refer to Table 4 in your text; if accepted, production will need this reference to link the reader to the Table.

Reviewers' comments:

Reviewer's Responses to Questions

**Comments to the Author**

1. Is the manuscript technically sound, and do the data support the conclusions?

Reviewer #1: Yes

Reviewer #2: Yes

2. Has the statistical analysis been performed appropriately and rigorously? 

Reviewer #1: Yes

Reviewer #2: Yes

3. Have the authors made all data underlying the findings in their manuscript fully available?

Reviewer #1: Yes

Reviewer #2: No

4. Is the manuscript presented in an intelligible fashion and written in standard English?

Reviewer #1: Yes

Reviewer #2: Yes

5. Review Comments to the Author

Reviewer #1: DKI is taking a more prominent position in MR imaging since its inception nearly two decades ago. The manuscript is about very interesting and hot topic in the neuroscience area that I believe this approach will be helpful for future early diagnosis of especially MS and other neural disorders. In my opinion, this paper is worth to be published.

Reviewer #2: Herein, the authors acquired a fast DKI sequence in patients with relapsing remitting multiple sclerosis, and found that lesions with different imaging characteristics on conventional sequences (FLAIR, T1w, contrast-enhanced T1w) demonstrated significant differences in mean kurtosis, which was not detected utilizing their DKI-derived mean diffusivity metric. The conclusions of the paper are that the fast DKI sequence could potentially be leveraged as a tool to further replace gadolinium-enhanced imaging. The merits of this paper are its easy to acquire sequence that is superior to conventional DWI/DTI-based methods, as well as the overall thorough methods used for image analysis. Although it is clear that further work (additional subjects) are needed to validate this strategy, it is nevertheless a worthwhile effort.

1. Rather unbalanced patient and control population – would be worth commenting on; did you test for normal distribution (i.e. Kolmogorov-Smirnov test)? Also, please comment on the uneven sex distribution between the MS and control groups.

2. In addition to dimethyl fumarate, were any MS patients on other MS medication (DMARDS)? Please comment.

3. With regards to your acquisition, did you consider utilizing your newer 1-9-9 compensated acquisition? It would be worth discussing why you opted for the 1-3-9 acquisition instead, as both are efficient acquisitions?

4. Did you obtain a reversed phase-encoding direction b0 image for topup correction? If not, please indicate this limitation.

5. In the methods section, what is “rice-floor adjustment” in your postprocessing step?

6. In the methods section 2.4, you are referring to four different areas of white matter damage (as many would also consider NAWM as you’ve listed).

7. When creating the NAWM mask, in subtracting out the CE-L, did you base it on the neuroradiologists’ segmented contrast enhancing masks, or the LST segmentations (which would be more encompassing of the lesion dimensions on FLAIR)? Please comment.

8. Is there a reason you did not specifically include the temporal lobe in your NAWM ROIs?

9. Is there a reason you did not calculate the lesion fractional anisotropy and instead opted to compare to MD? If this data is available, it would be reasonable to present it.

10. In the discussion section, I think you may be misrepresenting the performance of QSM “However, in a meta-analysis by Gupta et al. [7], only FA was a robust parameter to differentiate between enhancing and non-enhancing lesions in MS patients, implicating a need for additional imaging markers to evaluate white matter and lesional damage.“ The article itself stated that QSM had the best performance for discriminating between enhancing and non-enhancing lesions, albeit with limited data.

11. In the discussion section, as to why MK did not show difference between HC and NAWM, it could also be the result of specificity of the metric to pathologic changes within the NAWM (it does not discriminate between myelin loss and axonal loss) as well as potential uncharacterized differences between your subject groups compared to other studies. I would suggest bringing up these points in your discussion.

12. An important note is that enhancing lesions are not necessarily acute, as some older lesions can enhance as well (recurrent active demyelination). Please indicate if this was something you took into consideration in your analysis (reviewed multiple comparison studies to determine the actual approximate age of the lesions, particularly the enhancing ones).

13. It would be worthwhile to comment on alternative diffusion imaging schemes, such as NODDI, and SMT. See reference: Yu, F. et al. Imaging G-Ratio in Multiple Sclerosis Using High-Gradient Diffusion MRI and Macromolecular Tissue Volume. AJNR Am J Neuroradiol 40, 1871-1877, doi:10.3174/ajnr.A6283 (2019).

6. PLOS authors have the option to publish the peer review history of their article (what does this mean?). If published, this will include your full peer review and any attached files.

Reviewer #1: No

Reviewer #2: No

---

## [Author Response · Author response to Decision Letter 0]

12 Dec 2020

The word “gender” is used in the manuscript, although “sex” is also used. Usually, “sex” (the biological designation) is meant. “Gender” is the social construct and is rarely relevant in neurologic disease. Please revise the text to use “sex” rather than “gender” throughout.

Thank you for pointing this out. We substituted the “gender” by “sex” in our manuscript. 

It is stated that the DKI acquisition is very quick and that the post-processing requires only a few seconds. However, this does not account for other aspects of the pre-processing (e.g. use of eddy, co-registration, etc.) As you have included exact timings for the acquisition and Matlab processing, please include timings for these other vital steps as well.

It is true that the pre-processing steps, such as eddy correction, image co-registration, etc., also require a fair amount of time. The time it takes to perform these pre-processing steps are depending on two important factors: First, it depends on the image data you have acquired, and second, it also depends on your workstation and software version of FSL you use to perform these steps. Therefore, the runtime is quite variable. However, we added the approx.. runtime of the pre-processing steps it took us to the manuscript.

Is that Matlab code for the DKI processing being made publicly available as well?

A collection of processing scripts for fast DKI are available here: 

https://github.com/sunenj/Fast-diffusion-kurtosis-imaging-DKI

The authors SNJ and BH can also be contacted if help is needed with implementation. We have updated the manuscript with this information. 

Reviewer #2

Rather unbalanced patient and control population – would be worth commenting on; did you test for normal distribution (i.e. Kolmogorov-Smirnov test)? Also, please comment on the uneven sex distribution between the MS and control groups.

Thank you for pointing out this very important limitation. Due to the complex imaging protocol that was applied in this study we were not able due to timely and monetary reasons to include the same number of healthy controls that agreed to underwent the same imaging protocols. Therefore, we could only include a smaller age-matched control group that underwent the same scan protocol but we were not able to match for gender. It was of utmost importance that the healthy controls were age matched to our RRMS study group since age-related changes in MD and MK have been widely reported (Sexton et al. 2014, Benitez et al. 2018). Though sex differences in DTI parameters have been reported in certain brain structures (e.g. thalamus, cingulum, corpus callosum) in older populations, the impact of gender on DTI parameters in the white matter in younger populations is still ambiguous (Kennedy and Raz 2009, Sullivan et al. 2001). With a median age of 36.2 years in our RRMS patients and 34.8 years in our healthy control group respectively, we do not expect an age-related bias in our results. Furthermore, we like to point out that the focus of our study was not to investigate differences between MS patients and healthy controls but rather to identify differences within the MS lesions. Therefore, this study was not planned as a case-control-study and healthy controls were merely included to obtain a reference values for MK and MD. Still, we agree with reviewer’s comment and provided information and references to the limitation section of our manuscript. (“We were able to include only eleven healthy controls who were age-matched, but not matched for sex. It was of high importance to select an aged-matched control group since age can significantly influence MK and MD values within the white matter, while the impact of sex on DTI parameters in the white matter, especially in younger populations, is still controversial.[46-48] So far, studies focusing on sex-specific differences of DKI-derived indices in the white matter are missing, which has to be taken into consideration when interpreting our results. Further, we like to emphasize that this study was not a case-control-study to investigate differences between MS patients and healthy controls. Healthy controls were merely included to obtain a reference values for MK and MD.”)

In addition to dimethyl fumarate, were any MS patients on other MS medication (DMARDS)? Please comment.

Each patients was treated with dimethyl fumarate prior to this study. No other MS medication were administered.

With regards to your acquisition, did you consider utilizing your newer 1-9-9 compensated acquisition? It would be worth discussing why you opted for the 1-3-9 acquisition instead, as both are efficient acquisitions?

It is true that the 1-9-9 method is more robust to diffusion encoding imperfections than the 1-3-9 method used here. However, since all data were acquired using the same MR system our data quality assurance efforts ensure that gradient performance is good and consistent in the data used in this study. Therefore, the robustness offered by 1-3-9 (scan-rescan variability <3% in most areas) is sufficient for our study and the increased acquisition speed offered by 1-3-9 over 1-9-9 was given priority. 

Did you obtain a reversed phase-encoding direction b0 image for topup correction? If not, please indicate this limitation.

We did not obtain a reversed phase-encoding direction b0 image for topup correction. Therefore, we could not correct for susceptibility induced distortions. We used FSL’s epi_reg to register the b0 image to our structural T1 image and all image registrations were visually confirmed for quality control. We agree with the reviewer that this limitation has to be stated in the manuscript. We added: “Also, we did not obtain a reversed phase-encoding direction b=0 image and could not correct for susceptibility induced distortions. We used FSL’s epi_reg to register the b0 image to the T1w space and image co-registrations were checked carefully to exclude inaccurate registrations. Still, minor co-registration errors were possible.”

In the methods section, what is “rice-floor adjustment” in your postprocessing step?

This is a noise correction step which is needed because in the complex k-space data the noise is Gaussian but in the magnitude data the noise is instead follows a Rician distribution. The method proposed in the reference [25] by Koay et al. links the variance of the magnitude MR signal to the variance of the true Gaussian noise in the two quadrature channels. We have used this processing pipeline in numerous publications and prefer to omit these technical details and instead refer to the original method papers.

In the methods section 2.4, you are referring to four different areas of white matter damage (as many would also consider NAWM as you’ve listed).

We fully agree with the reviewer’s comment and changed the sentence accordingly “In MS patients, we defined four different areas of white matter damage, which included: normal-appearing white matter (NAWM), contrast-enhancing lesions (CE-L), T2/FLAIR-hyperintense lesions (FLAIR-L) and black holes (BH).”

When creating the NAWM mask, in subtracting out the CE-L, did you base it on the neuroradiologists’ segmented contrast enhancing masks, or the LST segmentations (which would be more encompassing of the lesion dimensions on FLAIR)? Please comment.

The NAWM mask was manually outlined by two experienced neuroradiologists in considerations of the LST segmentations to avoid partial volume errors from MS lesions. We added this important information to the manuscript. “ROIs were placed manually by two neuroradiologists (C.T. and T.F.) both with 5 years of experience in MS imaging and in consideration of the LST segmentations to avoid partial volume errors from MS lesions.”

Is there a reason you did not specifically include the temporal lobe in your NAWM ROIs?

We did not include the temporal lobe for placements of the NAWM ROIs due to the considerable amount of FLAIR lesions and “dirty white matter” adjacent to the inferior horn of the lateral ventricles in many patients. Therefore, we wanted to prevent avoid partial volume errors from MS lesions in this specific area.

Is there a reason you did not calculate the lesion fractional anisotropy and instead opted to compare to MD? If this data is available, it would be reasonable to present it.

We certainly agree with the reviewer that it would be highly interesting to include the fractional anisotropy (FA) of the lesions as well. Unfortunately, we are not able to provide FA with the fast kurtosis protocol used in this study. 

In the discussion section, I think you may be misrepresenting the performance of QSM “However, in a meta-analysis by Gupta et al. [7], only FA was a robust parameter to differentiate between enhancing and non-enhancing lesions in MS patients, implicating a need for additional imaging markers to evaluate white matter and lesional damage.“ The article itself stated that QSM had the best performance for discriminating between enhancing and non-enhancing lesions, albeit with limited data.

We agree with the reviewer’s comment that QSM indeed has shown very promising results in differentiating enhancing and non-enhancing lesions in MS (Zhang et al.; AJNR 2016). Further studies support the usefulness of QSM to evaluate MS lesions and have shown, that QSM increases during lesion formation. (Zhang et al.; Neuroimage Clin. 2018 Jan 28;18:143-148; Deh et al; J Magn Reson Imaging. 2018 Nov;48(5):1281-1287). We changed the sentence accordingly and added further references. “In a meta-analysis by Gupta et al. [7], only FA and QSM have shown reasonable accuracy measures to differentiate between enhancing and non-enhancing lesions in MS patients. Especially QSM seems to be a sensitive imaging marker to predict lesion enhancement.[30-32] However, studies focusing on potential and promising imaging markers in MS are still rare, implicating a need for additional quantitative measures to evaluate white matter and lesional damage.”

In the discussion section, as to why MK did not show difference between HC and NAWM, it could also be the result of specificity of the metric to pathologic changes within the NAWM (it does not discriminate between myelin loss and axonal loss) as well as potential uncharacterized differences between your subject groups compared to other studies. I would suggest bringing up these points in your discussion.

Thank you for pointing this out. Though we could not find a significant difference between HC and NAWM, we still found a trend towards decreased MK within NAWM (p=0.052). We also agree with the reviewer, that there are potential differences in our study cohort compared to other studies, who found significant differences in MK between HC and NAWM. In combination with the low specificity of MK to pathologic changes in NAWM, it might explain our results. We further emphasized this in our discussion. “Still, potential differences in our study cohort compared to the study cohorts from the studies mentioned above, such as the treatment with dimethyl fumarate or other uncharacterized parameters, might explain our results. Further, it has to be stated that MK is an empirical diffusion measure and offers low microstructural and pathological specificity to evaluate tissue damage in NAWM.[11]”

An important note is that enhancing lesions are not necessarily acute, as some older lesions can enhance as well (recurrent active demyelination). Please indicate if this was something you took into consideration in your analysis (reviewed multiple comparison studies to determine the actual approximate age of the lesions, particularly the enhancing ones).

This is a very interesting and important topic. Again, we agree with the reviewer that also “older” lesion can demonstrate enhancement, indicating recurrent active demyelination. Due to the low number of contrast-enhancing lesions in our study (n=30) and the missing discrimination between ring-like and nodular enhancement in the 2017 revisions of the McDonald criteria, we did not distinguish between ring-like and nodular enhancing lesions. It would be highly interesting to gain further insight into contrast-enhancement patterns using DKI. However, with the low number of contrast-enhancing lesions in our study cohort it will need further research to do that.

It would be worthwhile to comment on alternative diffusion imaging schemes, such as NODDI, and SMT. See reference: Yu, F. et al. Imaging G-Ratio in Multiple Sclerosis Using High-Gradient Diffusion MRI and Macromolecular Tissue Volume. AJNR Am J Neuroradiol 40, 1871-1877, doi:10.3174/ajnr.A6283 (2019).

Indeed, alternative diffusion imaging schemes have shown promising results and are important in detecting and evaluating demyelination and axonal damage in MS patients. We added this point to our discussion including the reference. It would be highly interesting to apply DKI in combination with other advanced quantitative MR sequences, such as MWI, QSM or alternative diffusion imaging schemes, such as neurite orientation dispersion and density imaging (NODDI) and spherical mean technique (SMT), which have shown very promising results to evaluate white lesional damage in MS patients.[18,32,58] Of course, histopathological correlations would provide the best evidence to verify our findings but are naturally limited.

---

## [Decision Letter · Decision Letter 1]

11 Jan 2021

Heterogeneity of Multiple Sclerosis Lesions in Fast Diffusional Kurtosis Imaging

PONE-D-20-30787R1

Dear Dr. Thaler,

We’re pleased to inform you that your manuscript has been judged scientifically suitable for publication and will be formally accepted for publication once it meets all outstanding technical requirements.

Kind regards,

Niels Bergsland

Academic Editor

PLOS ONE

Additional Editor Comments (optional):

Reviewers' comments:

Reviewer's Responses to Questions

**Comments to the Author**

1. If the authors have adequately addressed your comments raised in a previous round of review and you feel that this manuscript is now acceptable for publication, you may indicate that here to bypass the “Comments to the Author” section, enter your conflict of interest statement in the “Confidential to Editor” section, and submit your "Accept" recommendation.

Reviewer #2: All comments have been addressed

2. Is the manuscript technically sound, and do the data support the conclusions?

Reviewer #2: Yes

3. Has the statistical analysis been performed appropriately and rigorously? 

Reviewer #2: Yes

4. Have the authors made all data underlying the findings in their manuscript fully available?

Reviewer #2: Yes

5. Is the manuscript presented in an intelligible fashion and written in standard English?

Reviewer #2: Yes

6. Review Comments to the Author

Reviewer #2: The authors have sufficiently addressed most of the questions posed by the reviewers. It would seem the intrinsic limitations of this as a study performed in the clinical study intrinsically limits flexibility. With that said, the authors provided a thoughtful review of the comments and critiques posed. A fast diffusion kurtosis protocol intrinsically has appeal over conventional DTI, and this well-written paper shows a potential quantitative application in multiple sclerosis.

7. PLOS authors have the option to publish the peer review history of their article (what does this mean?). If published, this will include your full peer review and any attached files.

Reviewer #2: No

---

## [Editor Report · Acceptance letter]

14 Jan 2021

PONE-D-20-30787R1 

Heterogeneity of Multiple Sclerosis Lesions in Fast Diffusional Kurtosis Imaging 

Dear Dr. Thaler:

I'm pleased to inform you that your manuscript has been deemed suitable for publication in PLOS ONE. Congratulations! Your manuscript is now with our production department. 

Kind regards, 

on behalf of

Dr. Niels Bergsland 

Academic Editor

PLOS ONE